# Differential Expression of miRNAs Involved in Response to *Candidatus* Liberibacter asiaticus Infection in Mexican Lime at Early and Late Stages of Huanglongbing Disease

**DOI:** 10.3390/plants12051039

**Published:** 2023-02-24

**Authors:** Ana Marlenne Bojórquez-Orozco, Ángela Paulina Arce-Leal, Ricardo A. Chávez Montes, María Elena Santos-Cervantes, Abraham Cruz-Mendívil, Jesús Méndez-Lozano, Araceli G. Castillo, Edgar A. Rodríguez-Negrete, Norma Elena Leyva-López

**Affiliations:** 1Instituto Politécnico Nacional, CIIDIR Unidad Sinaloa, Departamento de Biotecnología Agrícola, Guasave 81101, Sinaloa, Mexico; 2Institute of Genomics for Crop Abiotic Stress Tolerance, Texas Tech University, Lubbock, TX 79409, USA; 3CONACYT—Instituto Politécnico Nacional, CIIDIR Unidad Sinaloa, Departamento de Biotecnología Agrícola, Guasave 81101, Sinaloa, Mexico; 4Instituto de Hortofruticultura Subtropical y Mediterránea “La Mayora” (IHSM), Universidad de Málaga-Consejo Superior de Investigaciones Científicas, Área de Genética, Facultad de Ciencias, E-29071 Málaga, Spain

**Keywords:** *Candidatus* Liberibacter asiaticus (*C*Las), Huanglongbing (HLB), Mexican lime, ShortStack, microRNA, target genes

## Abstract

Huanglongbing (HLB) is one of the most destructive diseases threatening citriculture worldwide. This disease has been associated with α-proteobacteria species, namely *Candidatus* Liberibacter. Due to the unculturable nature of the causal agent, it has been difficult to mitigate the disease, and nowadays a cure is not available. MicroRNAs (miRNAs) are key regulators of gene expression, playing an essential role in abiotic and biotic stress in plants including antibacterial responses. However, knowledge derived from non-model systems including *Candidatus* Liberibacter asiaticus (*C*Las)-citrus pathosystem remains largely unknown. In this study, small RNA profiles from Mexican lime (*Citrus aurantifolia*) plants infected with *C*Las at asymptomatic and symptomatic stages were generated by sRNA-Seq, and miRNAs were obtained with ShortStack software. A total of 46 miRNAs, including 29 known miRNAs and 17 novel miRNAs, were identified in Mexican lime. Among them, six miRNAs were deregulated in the asymptomatic stage, highlighting the up regulation of two new miRNAs. Meanwhile, eight miRNAs were differentially expressed in the symptomatic stage of the disease. The target genes of miRNAs were related to protein modification, transcription factors, and enzyme-coding genes. Our results provide new insights into miRNA-mediated regulation in *C. aurantifolia* in response to *C*Las infection. This information will be useful to understand molecular mechanisms behind the defense and pathogenesis of HLB.

## 1. Introduction

Huanglongbing (HLB), or citrus greening, is considered the most devastating and threatening disease in citriculture worldwide, causing a reduction in fruit quantity and quality. The causal agent is a phloem-limited, Gram-negative, α-proteobacterium belonging to the genus *Candidatus* Liberibacter. It is associated with three species, namely *Candidatus* Liberibacter asiaticus (*C*Las), *Ca*. L. africanus, and *Ca*. L. americanus [1,2]. The dominant form is associated with *C*Las, which is mainly transmitted by the Asian citrus psyllid, *Diaphorina citri* [3]. In Mexico, the first report of HLB was in 2009 in the state of Yucatan; since then, *C*Las has been detected in 24 of 28 citrus growing states [4]. Typical HLB symptoms include blotchy, mottled, and asymmetrical patterns in leaves, yellow shoots, and malformed fruits with aborted seeds [5,6].

Despite advances in research, there is no cure or tolerant citrus cultivars to HLB to date. To find tolerance genes, the first step requires basic knowledge of plant–pathogen interaction. The unculturable nature of *Candidatus* Liberibacter asiaticus limits its study. Notwithstanding, significant progress has been made in understanding proteomic and transcriptomic approaches in citrus [7,8,9,10,11,12,13,14]. Recently, our group identified several candidate genes involved in the Mexican lime–*C*Las interaction using a transcriptomic analysis. Differentially expressed genes were categorized into various groups of stress response at early and late stages of HLB. *C*Las-induced accumulation of genes related to amino acid, transport, Krebs cycle, hormone metabolism, and secondary metabolism. Meanwhile, transcriptional repression was mainly observed in redox, photosynthesis, and cell wall metabolic pathways [15]. 

MicroRNAs (miRNAs) have emerged as key regulatory molecules in response to abiotic and biotic stress. MiRNAs are a class of small non-coding RNA sequences with a usual range of 20 to 24 nucleotides in length. These molecules negatively regulate gene expression through cleavage or translation inhibition, playing an essential role in different biological and metabolic processes in plants [16,17,18].

MiRNAs are identified by experimental and computational analysis including direct cloning, next-generation deep sequencing (NGS), and available expressed sequence tags (ESTs) [19,20]. Traditionally, miRBase has been used to annotate miRNA sequences by homology search, and consequently, the number of miRNAs has increased over recent years. Nevertheless, the majority of identified miRNAs are doubtful; hence, it is necessary to emphasize the use of correct tools to find authentic miRNAs and minimize the population of false positives. ShortStack is a software that analyzes reference-aligned sRNA-Seq data and annotates miRNA sequences that are biologically representative [21,22,23]. ShortStack has been reported to be highly specific, and it has one of the lowest false positive rates, making it an efficient option for miRNA identification [22,24].

In plant–bacteria interaction, miRNAs are involved in two layers of the plant immune system, PAMP (pathogen-associated molecular patterns)-triggered immunity (PTI) and effector-triggered immunity (ETI). Some miRNAs, such as miR393 and miR160, belong to highly conserved miRNA families [25,26]. The expression patterns and functions of miR393 have been previously studied and Transport Inhibitor Response 1 (TIR1) and Auxin-Signaling F-box (AFBs) are auxin receptors and target genes of miR393. In *Arabidopsis*, repression of auxin signaling reduces *Pseudomonas syringae* pv. *tomato* DC3000 growth and enhances resistance against bacteria [27,28]. Likewise, overexpression of miR160a in transgenic *Arabidopsis* plants increases callose deposition by flagellin peptide (flg22) treatment of *Pseudomonas syringae* [29]. The regulatory network of these miRNAs indicates that they play an essential role in disease resistance.

The described roles of miRNAs in non-model plant–pathogen systems are limited. Regarding *C*Las infection in citrus, Zhao et al. [30] showed that miR399 is specifically upregulated in sweet orange (*Citrus sinensis*) leaves. The target gene on miR399 encodes a ubiquitin-conjugating enzyme (PHO2) related to degrading phosphorus transporter proteins. Additionally, treated plants with phosphorus oxyanion solutions showed significantly reduced HLB symptoms, suggesting that phosphorus deficiency is a critical factor in HLB symptomology. Recently, a total of 186 known and 71 novel miRNAs were identified from the roots of “Sanhu” tangerine (*Citrus reticulata*) infected by *C*Las, including cre-miR156a, cre-miR396b, cre-miR396g-5p, and cre-miRn70 as differentially expressed [31]. These authors expressed the need to generate more studies aimed to find HLB resistance. 

Despite the numerous studies on HLB and the socioeconomic value of Mexican lime, the miRNAs involved in *C*Las infection remain unknown. In this study, we identified known and novel miRNAs and their target genes at the early and late stages of HLB disease by NGS. These results facilitate the understanding of the role of miRNAs in the bipartite Mexican lime–*C*Las interaction to elucidate the molecular mechanisms, which will contribute to developing efficient alternatives for disease management.

## 2. Results

### 2.1. Data Analysis of Small RNA Sequencing 

Illumina sequencing generated eight small RNA libraries ranging from 34 to 54 million raw reads (Table 1). After removing adapters and low-quality reads, and size selection, a total of 22 to 37 million clean reads per library were obtained and used for further analysis. Length distribution analysis was performed in clean reads, and as expected, the most abundant reads were 24 nt in length representing 35.40% to 44.64% of the total clean reads, followed by 21, 22, and 23 nt (Figure 1). The eight sRNA libraries had a similar length distribution and showed no obvious differences between the asymptomatic (8 wpi) and symptomatic stage (16 wpi). This length distribution pattern is consistent with the biological nature of sRNAs in other plants. Finally, 48.86% to 55.75% of the clean reads were successfully mapped to the *Citrus aurantifolia* transcriptome.

### 2.2. Identification of Known and Novel miRNAs in Mexican Lime

ShortStack was used to identify miRNAs in *Citrus aurantifolia* by considering the fulfillment of the sixteen criteria (Appendix A). Clusters ranging from 5336 to 5956 sRNAs were identified in the eight libraries mapped to the *C. aurantifolia* transcriptome. According to ShortStack, approximately between 99.45% and 99.60% of the sRNA clusters were not determined as miRNAs (Appendix A). Nevertheless, ShortStack is a tool that follows rigorous criteria to determine miRNA annotation. To identify known miRNAs, all the sRNAs were aligned to the database of Plant small RNA genes (https://plantsmallrnagenes.science.psu.edu accessed on 24 February 2022), allowing no more than two mismatches. 

A total of 46 miRNAs from the eight citrus samples were identified by ShortStack, including 29 known miRNAs and 17 novel miRNAs, corresponding to 35 unique sequences (Table 2). The known miRNAs were classified into 15 families. MiR482 and miR164 were the most abundant families with four members each. MiR156, miR403, and miR391 had three members each and miR399 and miR166 contained two members; furthermore, eight families (miR159, miR172, miR167, miR396, miR162, miR393, miR168, and miR160) had only one member. The size of the majority of the mature miRNAs was 21 nt with 26 sequences, followed by 22, 24, and 20 nt with 12, 6, and 2 mature sequences, respectively. Previously, Axtell and Meyers [23] published the criteria for annotating novel miRNAs, and ShortStack follows those recommendations. According to the length, miRNAs of 23 or 24 nucleotides are scarce, requiring their accumulation in a minimum of four libraries. Here, three of the six 24-nt miRNAs fulfill these criteria. Even so, the minimum free energy (MFE) of the novel miRNA precursors were superior to −53.62 kcal/mol, except cau-miR008, suggesting a strong stability in their secondary structure. The normalized expression of miRNAs in reads per million (RPM) varied significantly, with a range from 0.598 to 69,297.24 (Appendix A). The known miRNAs showed a higher expression level compared with novel miRNAs. MiR166 and miR159 represented the top two families with a higher number of read counts, with more than 9681.477 RPM per library. Among novel miRNAs, cau-miR002, cau-miR003, and cau-miR009 showed more abundance, ranging from 128.21 to 633.544 RPM. A few miRNAs were identified in only one or two libraries, probably because some miRNAs were expressed at a specific stage of *C*Las infection or because of low expression levels.

Additionally, the remaining sRNA sequences were compared with the Plant small RNA genes database. We found 27 sRNA sequences that perfectly match with mature miRNAs in other plants (such as miR3951, miR393, and miR399); however, these sequences did not fulfill the criteria evaluated by ShortStack (Appendix A). Considering ShortStack is a stringent tool, these miRNAs could be false negatives, thus their biological validation and further studies are required. 

### 2.3. Differential Expression of miRNAs in Response to CLas in Citrus aurantifolia 

To identify miRNAs that respond to *C*Las infection, differential expression analysis was performed at the early (8 wpi) and late (16 wpi) stages of HLB disease using the read counts obtained by ShortStack. The results showed that 386 and 566 sRNA clusters were differentially expressed in the asymptomatic and symptomatic stages, respectively (Appendix A). Of the 46 miRNAs identified, at an early stage of the disease, four known (cau-miR403a, cau-miR403b, cau-miR403b, cau-miR172a) and two novel miRNAs (cau-miR006a and cau-miR011) were differentially expressed (Figure 2a). Additionally, miR3951 did not fulfill the criteria by ShortStack; however, it was identified as a miRNA in Plant small RNA genes and miRBase databases and was thus included for further analysis. 

On the other hand, eight known miRNAs were differentially expressed at the late stage, corresponding to cau-miR156a, cau-miR156b, cau-miR156c, cau-miR160a, cau-miR482a, cau-miR482c, cau-miR399a, and cau-miR399b. We include the sequences miR399, miR393, and miR3951 that were identified as miRNAs in databases (Figure 2b). Overall, three miRNAs were upregulated and four were downregulated during the early stage in response to *C*Las, whereas at the late stage, six miRNAs were upregulated and five were downregulated. Interestingly, during the initial stage of the disease, *C*Las induced the expression of the two novel miRNAs (cau-miR006a and cau-miR011), being two of the most upregulated miRNAs. Contrariwise, the miR399 family was the most downregulated, with a more than three-fold change. These miRNA expression levels suggested an important role of miRNAs in bacterial infections.

### 2.4. Prediction of Potential Target Genes of the Differentially Expressed miRNAs

To gain further insights into the biological role of miRNAs during *C*Las infection, we used the transcriptome of *C. aurantifolia* and the psRNATarget server to identify putative target genes. A total of 436 and 645 target genes were predicted from differentially expressed miRNAs for the asymptomatic and symptomatic stages, respectively. The top 10 candidate target genes based in expectation value of psRNATarget were selected for each miRNA. The seed region between miRNA and target was evaluated, allowing only one mismatch. The hybridization free energy in the asymptomatic stage ranged from −8.36 kcal/mol to −25.08 kcal/mol; meanwhile, in the symptomatic stage, it was −4.58 kcal/mol to −34.76 kcal/mol. Several genes showed unfavorable binding sites (Appendix A). This is because the energy required to open nucleotide bonds in the interaction was high, generating a decrease in free energy, causing less stability in miRNA-gene binding. According to the differential expression analysis of target genes obtained by Arce-Leal et al. [15], 16 and 44 genes were discarded for presenting the same expression pattern of their respective miRNA in the asymptomatic and symptomatic stages, respectively (Appendix A).

Consistent with previous studies, the target genes for most known miRNAs were transcriptional factors, signal transduction, and resistance proteins such as TIR-NBS-LRR (target of cau-miR482), involved in the first line in the detection of pathogens, including bacteria, and other proteins. The selected targets of the novel miRNAs were mainly related to the degradation, modification, and transport of proteins. A total of two target genes were chosen for each differentially expressed miRNA (Table 3).

### 2.5. Validation of miRNAs and Target Genes

To validate the expression of differentially expressed miRNAs due to *C*Las infection, five miRNAs and six target genes were evaluated by RT-qPCR. The consistency obtained in the expression levels of the selected miRNAs and target genes between the RT-qPCR and the RNA-seq data is shown in Figure 3. In both cases (symptomatic and asymptomatic), the expression pattern of the target genes was opposite to the corresponding miRNA. For example, cau-miR172a was downregulated with a log_2_ fold-change of −0.433 in the asymptomatic stage. The result showed by RT-qPCR was −0.389. Meanwhile, its target gene, Citrus_au_0000046354 (transcription factor MYB105), was induced with a log_2_ fold-change of 0.446 and 0.787 in sRNA-Seq and RT-qPCR, respectively. This confirms the reliability of the data analysis. 

## 3. Discussion

Huanglongbing (HLB), or greening disease, is the biggest threat to citriculture, and it spreads widely. When *C*Las is transmitted to the plant, the bacteria mainly replicate in the roots, while in foliar tissue, the titer is very low and cannot be detected during the first months of infection [32]. Nowadays, all citrus varieties are susceptible to the disease, leading to tree death a few years after infection [33,34]. Despite the efforts to study the disease and the impact of *C*Las in citrus crops, at present, there is no effective control or cure available. Recently, high-throughput sequencing technologies have developed rapidly, achieving the identification of known and novel transcripts with high sensitivity in plants [35,36]. Numerous miRNAs have been identified in many plants in response to pathogen attacks. Among the few studies related to miRNA identification in *C*Las infection [30,31], none of them studied the disease in *C. aurantifolia*, even though it is one of the most relevant citrus crops. In this study, we obtained sRNA-Seq libraries of *C. aurantifolia* at the asymptomatic (early, 8 wpi) and symptomatic (late, 16 wpi) stages of HLB development. These libraries were compared with their controls (HLB-) to detect miRNAs associated with *C*Las infection. The sequencing data showed that the most abundant sRNAs were 24 nt, followed by 21 nt. The lengths between 21 and 24 are consistent with the biological nature of sRNAs [37,38]. The size distributions agree with previous studies, such as *C. sinensis* in leaves [39], roots [40,41], and fruits [42]; *C. trifoliata* in fruits and flowers [43]; and *C. Junos* in roots [44]. Therefore, the sRNA-Seq data obtained here was reliable for further analysis.

More than 22 million filtered reads per library were obtained and then successfully aligned to Mexican lime transcriptome with a range of 48.86–55.77%. The identification of miRNAs by ShortStack provided relevant and reliable information of miRNAs annotation by reducing false positives and increasing precision and accuracy. Previously, the transcriptome of *Paspalum notatum* was used as a reference in ShortStack, obtaining a 30.3% reads alignment [45]. In addition, 24 miRNAs were detected in *L. campestre* small RNA data using the *L. campestre* genome; meanwhile, 29 miRNAs were identified in *L. appelianum* sRNA data using the *L. appelianum* transcriptome with ShortStack [46]. We detected 46 miRNAs in *C. aurantifolia*, corresponding to 29 known miRNAs and 17 novel miRNAs. Known miRNAs were identified in several plant species, being the most abundant sequences. Previous works have reported this association between the abundance of miRNAs and their conservation across the plant kingdom [37,47,48]. Our results show that cau-miR166, cau-miR159, and cau-miR396 were the most abundant families, similar to other works in citrus plants [30,49,50]. Of the novel miRNAs, most had low expression, including 12 miRNAs with a range of 0.598 to 11.627 RPM. In previous studies, novel sequences represent a small percentage of all miRNAs, and most of these miRNAs are present in low abundance [37,51,52].

The differentially expressed miRNAs at the asymptomatic stage showed the repression of cau-miR403 and cau-miR172. MiR403 has been reported in soybean plants (*Glycine max*) infected with the oomycete *Phytophthora sojae* [53]. Similarly, Zhang et al. [25] reported miR403 repression in *Arabidopsis* plants infected with three different strains of *Pseudomonas syringae* (Pst DC3000 EV, Pst DC3000 hrcC, Pst DC3000 avrRpt2), suggesting a potential role in plant immunity. This is because the target gene of miR403 is Argonaute 2 (AGO2), in association with AGO1 [54]. The regulatory network miR403/AGO2/AGO1 has been validated with viruses, where repression of AGO1 by viral suppressors leads to negative regulation of miR403 and subsequent induction of AGO2 [55]. However, AGO2 induction by bacteria could be mechanistically different from virus-triggered induction [25]. Hence, further studies are required to elucidate the relationship between miR403 and bacterial infection in plants. In our work, AGO2 was one of the targets with the best rate. Nevertheless, the gene had the same expression pattern as the miRNA for which it is a target, thus it was discarded. Previously, miR172 has been shown to play an important role in plant growth and development [56]. Overexpression of this miRNA disrupts normal leaf and flower development [57], associating miR172 with the occurrence of disease symptoms in plants [58]. The target gene of miR172 is MYB105. MYB genes play an important function in plant defense against pathogens and plant hormone responses [59]. A previous work reported that MYB gene expression contributed to defense against *Pyricularia oryzae* and *Xanthomonas oryzae* in rice [60].

Among novel miRNAs, cau-miR011 and cau-miR006 represent the upregulated miRNAs in the early stage of infection. Cau-miR011 represses the expression of SUC3 gene, a protein responsible for sucrose transport in plants. Like most fruit species, sucrose is the main photoassimilate that is synthesized in leaves and then exported through the phloem sieve elements to sink tissue in citrus [61,62]. In a previous study, Fan et al. [63] reported accumulated sucrose in leaves of *C. sinensis* infected by *C*Las, concluding that photoassimilate translocation is affected by HLB infection. On the other hand, Martinelli et al. [64] argued that sucrose transporter genes were induced in peel tissue in symptomatic fruits (HLB+). Likewise, sucrose also acts as a signaling compound that can alter gene expression in plant growth and development [65]. It has been well-established that HLB affects carbohydrate metabolism, which suggests that cau-miR011, by regulating the expression of sucrose transporter proteins and inhibiting their action, could have generated the accumulation of sucrose in *C*Las-infected leaves, resulting in the impairment of photoassimilate translocation. This could contribute to small fruits and physiological disorders in plants, which are typical symptoms of HLB. However, it is necessary to study the biological role of cau-miR011. Cau-miR006 regulates the expression of the FtsH extracellular protease family. This target is an ATP-dependent zinc metalloprotease involved in plant response to stress, maintaining a normal progress and stability in the chloroplast [66]. Loss of FtsH leads to different phenotypes, including pale seedlings, albino seeds, ROS accumulation, and disrupted thylakoid formation [67,68,69].

Regarding miRNAs differentially expressed at the symptomatic stage (16 wpi), the same expression pattern of miR393 and miR160 has already been described in previous works in *C. sinensis* infected with *C*Las [30] and *Arabidopsis*–*Pseudomonas* interaction [29,70]. Interestingly, in our study, cau-miR399 was the most downregulated miRNA and is considered a specific miRNA of HLB infection. Contrastingly, miR399 was significantly upregulated in *C. sinensis*, causing the degradation of mRNA encoding UBC24 (pho2) enzyme responsible for phosphorus translocation and remobilization [30]. MiR399 induces its expression in response to phosphorus starvation in HLB-positive samples; however, in our study, miR399 was highly downregulated in the symptomatic stage. Additionally, its target gene (UBC24) was differentially expressed with a 1.277 log_2_ fold-change, suggesting that miR399 response is differential among citrus cultivars.

It is known that the ubiquitin-proteosome system is the main protein turnover pathway responsible for protein degradation and modification in eukaryotic cells, contributing to many aspects of cellular processes [71]. Ubiquitination is a process that requires ubiquitin-activating enzyme (E1), ubiquitin-conjugating enzyme (E2), and ubiquitin ligase (E3). Specifically, E2 regulates the interaction, function, and destination of target genes [72,73]. Cau-miR403 induces the expression of ubiquitin-conjugating enzyme E2 (UBC38). Likewise, cau-miR399 induced the expression of ubiquitin-conjugating enzyme E2 (UBC24). This ubiquitin has been described in previous works [30,74,75,76]. Ubiquitin E3 are classified in specific domains such as U-box, specifically PUB40 degraded BZR1; the functions are related to biotic and abiotic stress [77]. Cau-miR006 downregulated the expression of PUB40. Despite the importance of E2 and E3 enzymes, information and studies are largely lacking [78].

The first defense mechanism of plants is based on the initial recognition of pathogen by pattern recognition receptors (PRRs) receiving signals about the attack to activate the immune response denominated PAMP-triggered immunity (PTI) through protein interactions. Important signaling factors involve mitogen-activated protein kinase (MAPK) cascades [79,80]. The novel miRNA cau-miR011 regulated MAPKKK10 in the early stage of the disease. In addition, the plant hormone auxin is known to be involved in many plant processes; the repression of auxin increases plant resistance to pathogens [27]. Cau-miR160 regulated the expression of ARF17, playing an important role in the infection and suggesting that suppression of ARF could lead to a partial resistance to the bacteria. A second pathway to detect pathogen attack is through effector recognition, leading to the activation of effector-triggered immunity (ETI). MiRNAs regulate hundreds of disease resistance genes (R-genes) by targeting sites with similar nucleotide motifs [81]. Among the predicted target genes, one of the most reported gene was LRR (miR3951), including TIR-NBS-LRR (miR482). In a previous work, the genes LRR1 and LRR2 were induced in tomato plants during infection by *Pst* DC3000, while the expression of miR482 was downregulated [82]. Our results showed the downregulation of R-genes in both stages of HLB disease. 

At the symptomatic stage, cau-miR156a and cau-miR156b were downregulated, while cau-miR156c was upregulated. Precursors and mature miRNA sequences are different. However, the miRNAs share the same family of target genes, corresponding to Squamosa promoter-binding-like protein (SPL). SPL plays an important role in plant development of flowers and fruits. The overexpression of miR156 in *Arabidopsis* infected with *Pst* DC3000 induced susceptibility to the bacteria, whereas overexpression of SPL9 increased resistance in the plant [83]. Similarly, a recent report demonstrated that overexpression of OsSPL4 in transgenic rice plants enhanced disease resistance against *Magnaporthe oryzae* [84].

In the present study we generated a homogeneous and accurate experimental *C. aurantifolia*-CLas system, by selecting 8 and 16 wpi as asymptomatic and early symptomatic points with the idea to avoid the pleiotropic effects observed in very late infection stages. In both asymptomatic and symptomatic stages, all detected miRNAs are present; however, in order to understand the miRNAs-mediated response during infection, we analyzed only the differential expressed miRNAs. For this reason, only a few of them are common in both stages. Our data are in agreement with a previous study of miRNA differential response in a *C. sinensis*-CLas system, where not all detected miRNAs are differentially expressed during different disease stages [30]. Our results provide putative functions of miRNAs in response to *C*Las infection. This information would allow the development of sustainable management strategies, since in the end, the most desirable strategy for disease control in plants will always be the use of natural defense mechanisms.

## 4. Materials and Methods

### 4.1. Plant Material and Experimental Design

Mexican lime plants on alemow (*C. macrophylla*) rootstock (nine months after grafted) placed in 40 L pots with 20 kg substrate (Coconut coir dust, vermicompost, and perlite, 1:1:1) were used in this study. Plants were irrigated two or three times per week and fertilized every three weeks with a water-soluble fertilizer mixture (20-0-30 N-P-K) and micronutrients (Microfol^®^ Combi P.S., Biolchim, Bologna, Italy). Inoculation was carried out in February 2018. The inoculum source was symptomatic budwoods with the bacterium *Candidatus* Liberibacter asiaticus. The experimental design was carried out as previously described by Arce-Leal et al. [15]. A total of 45 plants were inoculated with budwood from *C*Las-infected Mexican lime trees, and 15 plants were inoculated with *C*Las-free budwood as controls. Plants were maintained in a 250 m² size shaded greenhouse with an anti-aphid insect screen at the INIFAP Experimental Station in Tecomán, Colima, Mexico, with an average annual temperature of 26 °C (range 16–34 °C) and a mean humidity of 64% (range 46–100%).

For this study, HLB disease progress was designated as early or late stage based on bacterial titer and plant symptoms. The early (asymptomatic) stage was designated for plants at eight weeks post-inoculation (8 wpi) of *C*Las and containing a homogeneous bacterial titer of ~2.5 × 10^2^ bacterial cells/100 ng of total DNA. The late stage (symptomatic) of HLB disease progress was considered at 16 wpi because plant symptoms at this point included leaf yellowing and asymmetric blotchy mottling [15] and a titer of ~1.4 × 10^4^ bacterial cells/100 ng of total DNA. Among 45 infected plants, 10 plants with *C*Las (HLB+) were selected according to homogeneity on the disease symptoms and bacterial titers. Among 15 healthy plants, 10 plants were randomly selected as controls (HLB-). In all selected plants, four complete leaves located at the third level of ramification from the main stem was collected at 8 and 16 wpi. The plant tissue was frozen with liquid nitrogen and ground with mortar and pestle. The same 10 *C*Las-infected plants and 10 control plants selected at 8 wpi were sampled at 16 wpi. 

### 4.2. sRNA Library Construction and Sequencing

Total RNA was extracted from the full leaves of the selected plants using the TRIzol^®^ protocol (Thermo Fisher Scientific, Carlsbad, CA, USA). The concentration and integrity of RNA samples were verified using a Nanodrop 2000 and capillary electrophoresis by the 2100 Bioanalyzer RNA Nano Chip (Agilent Technologies, Inc., Santa Clara, CA, USA). For each condition, total RNA from the 10 selected plants were divided in two groups, each one containing five plants and pooled in an equimolar ratio to construct each cDNA library. A total of eight small RNA libraries were generated, four at the early (two *C*Las inoculated and two negative controls) and four at the late (two *C*Las inoculated and two negative controls) stage of HLB disease progress. We generated libraries LM8wpiHLB + 1 and LM8wpiHLB + 2, each one containing five pooled 8 wpi asymptomatic plants, and libraries LM16wpiHLB + 5 and LM16wpiHLB + 6, each one containing five pooled 16 wpi symptomatic plants. Control libraries, LM8wpiHLB − 3 and LM8wpiHLB − 4 (for 8 wpi stage), and LM16wpiHLB − 7 and LM16wpiHLB − 8 (for 16 wpi stage), were constructed by pooling five mock-inoculated plants, respectively. Small RNA libraries were constructed using the TruSeq small RNA Library Prep Kit v2 Sample Preparation Kit (Illumina, San Diego, CA, USA) according to the manufacturer’s instructions. Libraries were sequenced (1 × 35 pb) on an Illumina NexSeq500 platform in Langebio-CINVESTAV, Irapuato (Guanajuato, Mexico) facilities. The plant tissue used for RNA-Seq analysis was the same as that used for RT-qPCR.

### 4.3. Identification of Known and Novel miRNAs in Mexican Lime

Raw data were processed using Atropos v1.1.28 [85] to remove adaptor sequences, low-quality reads, and contaminated reads. Reads smaller than 18 nt or longer than 27 nt were also removed. The quality control standards were evaluated with FASTQC software (http://www.bioinformatics.babraham.ac.uk/projects/fastqc/, accessed on 17 January 2022). Because the genome of *Citrus aurantifolia* is not available, clean reads were mapped (up to one mismatch) to the reference transcriptome [86] using Bowtie v1.2.3 software [87]. ShortStack v3.8.5 package (https://github.com/MikeAxtell/ShortStack, accessed on 21 January 2022) [21] was used to annotate and quantify the sRNAs previously aligned to the reference. Then, the sRNA sequences were aligned against the databases of Plant small RNA genes (https://plantsmallrnagenes.science.psu.edu, accessed on 24 February 2022) [88] and miRBase (https://www.mirbase.org, accessed on 7 March 2022), with no more than two mismatches to identify known mature miRNAs. For the known miRNAs, the miRNA family was assigned by considering the best match with the database of Plant small RNA genes, but with a new letter suffix to differentiate miRNA sequences. The names of novel miRNAs were assigned sequentially. 

### 4.4. Differential Expression Analysis of miRNAs 

Shortstack-derived read counts were used to identify differentially expressed miRNAs. The expression of sRNAs in each library was normalized and analyzed by DESeq2 v1.36 package (https://bioconductor.org/packages/release/bioc/html/DESeq2.html, accessed on 15 August 2022) [89]. Normalized data in asymptomatic and symptomatic samples were compared with their corresponding control samples. Those miRNAs with an adjusted *p*-value ≤ 0.05 were considered as differentially expressed.

### 4.5. Prediction and Annotation of Target Genes

The psRNATarget server (https://www.zhaolab.org/psRNATarget/analysis, accessed on 22 August 2022) [90] was used to predict the potential target genes of the differentially expressed miRNAs at early and late stages using the default parameters and a maximum expectation value of four. The transcriptome of *C. aurantifolia* [86] was used as a reference for the target search. The top 10 rate targets were chosen, and the criteria used to select the best two targets for each miRNA were as follows: (1) complementarity in the seed region (2–13 nt), (2) the binding properties miRNA-target with RNAUp server (http://rna.tbi.univie.ac.at/cgi-bin/RNAWebSuite/RNAup.cgi, accessed on 21 September 2022) [91], and (3) mRNA expression obtained by RNA-Seq [15]. Putative functions of the final targets were annotated in the NR protein database using BLASTx with default settings. 

### 4.6. Validation of miRNAs and Target Genes Expression by RT-qPCR

To validate the expression of differentially expressed miRNAs and target genes, stem-loop RT-qPCR [92] and RT-qPCR were performed, respectively. Five candidate miRNAs (two from early stage and three from late stage) and six target genes were selected. Total RNA (one microgram) from leaves of five plants was pooled in equimolar concentration to obtain eight samples, and M-MLV Reverse Transcriptase (Invitrogen) used to produce the cDNA according to the manufacturer’s instructions. Each condition was represented by two biological replicates. 

Specific primers for the miRNAs and target genes (Appendix A) were designed using Primer Select software (DNASTAR Lasergene, Madison, WI). qPCR reactions were performed on a CFX96TM real-time PCR system (Bio-Rad) and SsoFast EvaGreen^®^ Supermix (Bio-Rad, Foster City, CA, USA) according to the manufacturer’s instructions. The cycling program included incubation at 95 °C for 30 s followed by 40 cycles of denaturation at 95 °C for 5 s and annealing at 58 °C for 10 s. All reactions were analyzed using two technical replicates. Relative expression levels were calculated using the 2^−ΔΔCt^ method [93]. COX [94] and U6 [95] genes were used as internal controls for the target genes and the miRNAs, respectively.

## Figures and Tables

**Figure 1 plants-12-01039-f001:**
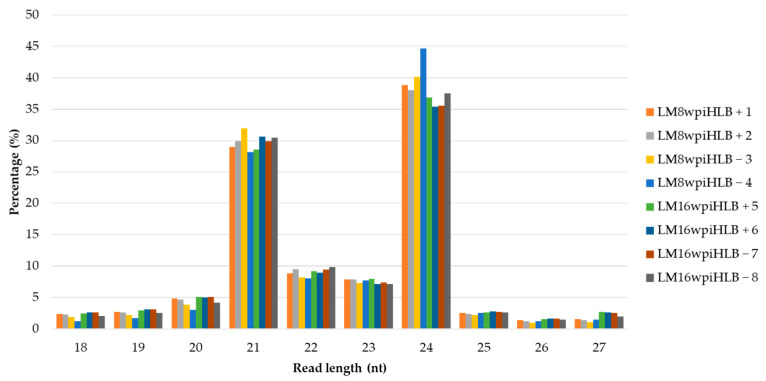
Small RNAs length distribution in libraries from asymptomatic and symptomatic stages of Huanglongbing (HLB) disease. HLB-infected samples (HLB + 1, 2, 5 and 6), HLB non-infected control samples (HLB − 3, 4, 7 and 8).

**Figure 2 plants-12-01039-f002:**
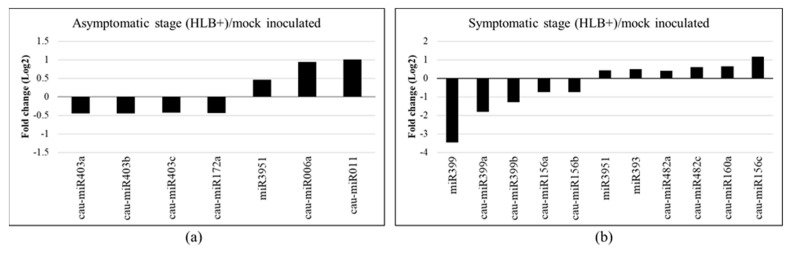
Differential expression of known and novel miRNAs in *Citrus aurantifolia* inoculated with *Candidatus* Liberibacter asiaticus (*C*Las). (**a**) Asymptomatic (8 wpi) and (**b**) Symptomatic (16 wpi) stages of HLB disease. Bars represent fold change (Log_2_) expression from HLB+/HLB− ratio of two biological replicates using read counts from ShortStack data.

**Figure 3 plants-12-01039-f003:**
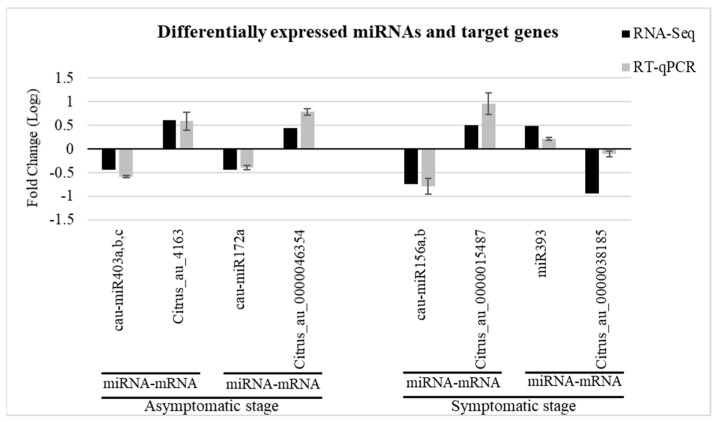
Experimental validation of the differentially expressed miRNAs and target genes by RT-qPCR. For RT-qPCR quantification, each bar represents relative expression levels calculated by 2^−ΔΔCt^ method. Error bars represent standard deviation of two biological replicates.

**Table 1 plants-12-01039-t001:** Data summary of sRNA-Seq libraries in *Citrus aurantifolia* at early and late stages of Huanglongbing disease.

Condition	Libraries	Raw Reads	Clean Reads	Reads Mapped to Transcriptome	miRNAs Identified	Total miRNAs
Asymptomatic (early) stage	LM8wpiHLB + 1	38,662,026	23,579,913	12,691,904 (53.82%)	24	46
	LM8wpiHLB + 2	48,578,991	29,405,251	16,123,718 (54.83%)	28	
Mock inoculated	LM8wpiHLB − 3	34,095,266	22,307,669	11,744,853 (52.64%)	29	
	LM8wpiHLB − 4	54,610,772	37,857,718	18,498,221 (48.86%)	27	
Symptomatic (late) stage	LM16wpiHLB + 5	47,150,743	25,075,202	13,584,848 (54.17%)	23	
	LM16wpiHLB + 6	41,940,661	22,189,688	12,312,553 (55.48%)	24	
Mock inoculated	LM16wpiHLB − 7	51,599,735	26,326,495	14,677,662 (55.75%)	23	
	LM16wpiHLB − 8	50,672,438	30,719,833	16,774,553 (54.60%)	26	

**Table 2 plants-12-01039-t002:** Known and novel miRNA sequences identified in Mexican lime.

	miRNA ^1^	Sequence (5′ to 3′)	Length	miRNA Location in *C. aurantifolia* Transcriptome
Known miRNAs	cau-miR156a,b	UUGACGGAAGAUAGAGAGCAC	21	Citrus_au_5549, Citrus_au_5550
	cau-miR156c	AUGACAGAAGAGAGAGAGUAC	21	Citrus_au_0000117891
	cau-miR159a	UUUGGAUUGAAGGGAGCUCUA	21	Citrus_au_0000113744
	cau-miR160a	UGCCUGGCUCCCUGUAUGCCA	21	Citrus_au_3426
	cau-miR162a	UCGAUAAACCUCUGCAUCCAG	21	Citrus_au_0000053652
	cau-miR164a,b	UGGAGAAGCAGGGCACGUGCA	21	Citrus_au_0000038181, Citrus_au_4480
	cau-miR164c,d	CAUGUGCCCUUCUUCCCCAUC	21	Citrus_au_4480, Citrus_au_0000038181
	cau-miR166a	UCGGACCAGGCUUCAUUCCCU	21	Citrus_au_3052
	cau-miR166b	UCGGACCAGGCUUCAUUCCCC	21	Citrus_au_7116
	cau-miR167a	UGAAGCUGCCAGCAUGAUCUGA	22	Citrus_au_7532
	cau-miR168a	UCGCUUGGUGCAGGUCGGGAA	21	Citrus_au_0000028247
	cau-miR172a	GUAGCAUCAUCAAGAUUCAC	20	Citrus_au_1099
	cau-miR391a	UGCAGGUGAGAUGAUACCGUCA	22	Citrus_au_9968
	cau-miR391b,c	CCGGAAUCAUUUCUCCCGCGUG	22	Citrus_au_0000052759, Citrus_au_9968
	cau-miR393a	UCCAAAGGGAUCGCAUUGAUCU	22	Citrus_au_0000052858
	cau-miR396a	UUCCACAGCUUUCUUGAACUG	21	Citrus_au_0000051770
	cau-miR399a	UGCCAAAGGAGAGUUGCCCUA	21	Citrus_au_6097
	cau-miR399b	UGCCAAAGGAGAGUUGCCCUG	21	Citrus_au_0000054379
	cau-miR403a,b,c	UUAGAUUCACGCACAAACUCG	21	Citrus_au_4163, Citrus_au_0000097776, Citrus_au_0000097777
	cau-miR482a,b,c	UUUUUCCCACACCUCCCAUCCC	22	Citrus_au_4020, Citrus_au_0000120512, Citrus_au_0000054570
	cau-miR482d	UCUUGCCCACCCCUCCCAUUCC	22	Citrus_au_7459
Novel miRNAs	cau-miR001	AUGUUGCUUGAUGAUAUUUAGUGU	24	Citrus_au_0000073759
	cau-miR002	CCGUUUCAUCUUGUCCUCCAG	21	Citrus_au_0000096565
	cau-miR003	CUGUAGAAGGCUCCUGUGACC	21	Citrus_au_5642
	cau-miR004	AAAGUUAGGGAUAAGUUAAAAGAC	24	Citrus_au_0000021657
	cau-miR005	UCUAAGAAAACUUCAAUAGCU	21	Citrus_au_8635
	cau-miR006a,b	UAGCAGCUGGGCUGUGAUAGGCCA	24	Citrus_au_0000043483, Citrus_au_5236
	cau-miR007	AGUUUCAGGAUUGGUUUGGGAUUC	24	Citrus_au_0000074192
	cau-miR008	GUGUUGCUUGAUGAUAUUUAGUGU	24	Citrus_au_0000073759
	cau-miR009	UCGCAGGAGCUUUCUACGGUU	21	Citrus_au_5642
	cau-miR010	AAUGUUGAGCUAAGGUUUUGC	21	Citrus_au_3278
	cau-miR011	UGACUCGUGAGUUGGCCUUG	20	Citrus_au_2177
	cau-miR012a,b,c	UUUUGUUGCAUGAUGCUGAUAA	22	Citrus_au_0000041993, Citrus_au_0000105499, Citrus_au_0000041992
	cau-miR013	AAAACCUCAACUCAGCACUGA	21	Citrus_au_3278
	cau-miR014	CUUUCAGCAGCCUUCGGCGUC	21	Citrus_au_0000116384

^1^ The different letters in suffix (a, b, c, d) indicates identical sequences or isoforms from the same family.

**Table 3 plants-12-01039-t003:** Differential expression of miRNAs and their target genes in asymptomatic (early) and symptomatic (late) stages of Huanglongbing (HLB) disease.

Stage	miRNA ^1^	miRNA Differential Expression (Log_2_ FC)	mRNA Differential Expression (Log_2_ FC)	Target Gene	Predicted Function	Inhibition
8 wpi	cau-miR403a,b,c	−0.445 −0.449 −0.430	0.609	Citrus_au_4163	Putative ubiquitin-conjugating enzyme E2 38	Cleavage
			0.714	Citrus_au_0000009992	Flavin-containing monooxygenase FMO GS-OX-like 2	Translation
	cau-miR006a	0.939	−0.603	Citrus_au_0000110774	FtsH extracellular protease family	Cleavage
			−1.248	Citrus_au_0000046578	U-box domain-containing protein 40 (PUB40)	Cleavage
	cau-miR011	1.010	−1.177	Citrus_au_0000014317	Sucrose transport protein SUC3	Cleavage
			−1.290	Citrus_au_2600	MAPK/ERK kinase kinase E493 (MAPKKK10)	Cleavage
	cau-miR172a	−0.433	0.446	Citrus_au_0000046354	Myb domain protein 105 (MYB105)	Cleavage
			0.491	Citrus_au_0000042060	Replication protein A 70 kDa DNA-binding subunit C (RPA1C)	Cleavage
	miR3951	0.467	−0.642	Citrus_au_4749	Leucine-rich repeat (LRR) family protein	Cleavage
			−0.762	Citrus_au_0000103503	Leucine-rich repeat (LRR) family protein	Cleavage
16 wpi	cau-miR156a,b	−0.746 −0.748	0.883	Citrus_au_0000112798	Squamosa promoter-binding-like protein 3	Cleavage
			0.504	Citrus_au_0000015487	Squamosa promoter-binding-like protein 4	Cleavage
	cau-miR156c	1.160	−1.247	Citrus_au_0000049123	Squamosa promoter-binding-like protein 4	Cleavage
			−2.045	Citrus_au_3104	UDP-glycosyltransferase 73C3	Cleavage
	cau-miR160a	0.649	−1.109	Citrus_au_0000004007	Auxin response factor 17 (ARF17)	Cleavage
			−0.554	Citrus_au_0000004009	Leucine aminopeptidase 3, chloroplastic	Cleavage
	cau-miR399a,b	−1.806 −1.291	1.277	Citrus_au_0000032113	Probable ubiquitin-conjugating enzyme E2 24 (UBC24)	Cleavage
			1.371	Citrus_au_0000046079	Dicarboxylate transporter 1, chloroplastic (DIT1)	Translation
	cau-miR482a,c	0.500	−0.444	Citrus_au_10466	Disease resistance protein (TIR-NBS-LRR class) family	Translation
			−0.509	Citrus_au_5033	hydrolases, acting on ester bonds	Cleavage
	miR393	0.487	−0.392	Citrus_au_0000018820	NADPH--cytochrome P450 reductase 1 (ATR1)	Cleavage
			−0.944	Citrus_au_0000038185	Amino acid kinase family protein	Cleavage
	miR3951	0.418	−0.878	Citrus_au_4749	Leucine-rich repeat (LRR) family protein	Cleavage
			−0.504	Citrus_au_0000041265	Calcium-dependent protein kinase 21 (CPK21)	Cleavage
	miR399	−3.455	1.277	Citrus_au_0000032113	Probable ubiquitin-conjugating enzyme E2 24 (UBC24)	Cleavage
			1.073	Citrus_au_0000042988	C2H2-like zinc finger protein	Cleavage

^1^ The different letters in suffix (a, b, c) indicates identical sequences or isoforms from the same family.

## Data Availability

The raw data presented in this study are openly available in NCBI database, reference number PRJNA574168.

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
