# Peer review of "Differential Expression of miRNAs Involved in Response to Candidatus Liberibacter asiaticus Infection in Mexican Lime at Early and Late Stages of Huanglongbing Disease"

_plants, 2023, doi:10.3390/plants12051039_

Round 1
Reviewer 1 Report
This is a review of the paper entitled “Differential expression of genuine miRNAs involved in response to Candidatus Liberibacter asiaticus infection in Mexican lime at early and late stages of Huanglongbing disease.” The authors looked at microRNAs in limes for possible keys that could be developed into therapeutic products for disease management. In this method they compared infected and healthy trees at two times (8 and 16-weeks post graft inoculation). While the idea was good, there are significant problems in execution. If the response to infection is a continuous function, then it is hard to see that function looking at the two time points that were sampled because there was only one miRNA in common (Fig 2). While the agreement between the two replicates (Figure 3) was encouraging, that does not tell a biological story. There should be a seasonal component, and there should be a cumulative component where damage at earlier stages of disease progression accumulate and result in secondary effects. Things like restricting the vascular system results in nutrient deficiency symptoms as a secondary effect. So, the experimental design feels like preliminary data, or proof of concept that there are differences. We now need confirmatory studies where the design is repeated over time, or the experiment is repeated with more intervals (2 weeks rather than 2 months). There were controls, but that assumes that the disease-plant interaction is constant over season. The authors could address that issue partly by comparing the two controls. To be more than slightly interesting I need to see how Figure 2a changes into Figure 2b, and I need to see how other factors, like season (daylength, temperature, light), influence the relationship. The use of ShortStack should facilitate identification of the strongest candidates, but programs are never 100% accurate for a variety of reasons not least of which is finding an ideal balance between false-positives and false-negatives. I am uncomfortable with apparently random outcomes. Links to other articles showing similar up- or down-regulation given a similar stimulus help, but only a little. Another few treatments would have greatly strengthened the outcome from this research by enabling the identification of miRNAs that follow a pattern rather than relying exclusively on ShortStack.
Title) “genuine” suggest that there are other options. Line 67-68 indicates many are doubtful and some are authentic. The problem appears to be false positives in the algorithms used to identify miRNAs. One way to get a very “accurate” program is to change the balance between false positives and false negatives. If one never checks the false-negative rate, the appearance is one of progress (as in line 72). What is the false negative rate for ShortStack?
Line 19) I would spell out spp to avoid a double period: spp. (for the abbreviation) and another period for the end of the sentence.
Line 20) a cure
Line 74) define acronyms at first use: PAMP. If I must look it up, I first find that PAMP are gold bars available through fine jewelers or investment firms. While the acronym may be “common” or maybe reviewers have suggested “just use the acronym” I suggest that a few years from now some of the acronyms will have a new “common” use and this paper becomes harder to read. You avoid all those problems if they are defined at first use.
Line 76) What are acronyms TIRI and AFB? Please try to minimize use of acronyms.
Line 125) Such programs have an error rate even if it is small. There is no guarantee that these are genuine. At best there is a lower error rate relative to another program had could have been used. However, that is not the same as zero error.
Line 131) “corresponding” does not make sense. End the sentence after families and start a new sentence.
Line 150) how is conserved the opposite of novel? Line 143 makes a comparison “known” versus “novel” and that makes sense. Use known rather than conserved.
Line 153) You tested against the database of plant small RNA genes. What about testing against Candidatus Liberibacter asiaticus genome, or is it not possible for CLas to have miRNA sequences?
Line 154) Are you sure Plant should be capitalized? Is this an in-house library, or is there a URL or proper citation?
Line 169) Bars are the (means?) of two replicates. The graph hides the variability in the data. As a ratio of HLB+ and HLB- it is not clear how the ratio was calculated. Are all possible pairs of HLB+ (n=2) and HLB- (n=2) used? Is there some “natural” pairing?
Line 240) The designation of early and late is an artifact of your experimental set-up. At least the trees I work with may not start symptom expression for years and symptom expression is irregular: one tree has it its neighbor does not. Also, the limes I worked with got HLB at the same time as some neighboring orange and grapefruit. Over the course of two years all the orange and grapefruit died. The limes were not “healthy” but only 1% were dead. Do you also see this? How might that influence interpreting the results?
Line 364) What is a shadow greenhouse? Shade?
Line 367) How were the plants selected? Were some plants in both 8- and 16-weeks?
Line 368) How do I select two groups of five to get 8 samples?
Line 371) Some methods are missing. Please describe plant age, soil, pot size, fertilization, watering, and pest management.
Ideally the authors should redo the experiment in some way. If nothing else it would show that the results are or are not repeatable.
Author Response
"Please see the attachment"

Reviewer 2 Report
Generally, the manuscript is logically organized and easy to follow. Therefore, I only have a few suggestions for the authors.
Major:
1, The conclusion “In both cases (symptomatic and asymptomatic), the expression pattern of the target genes was opposite to the corresponding miRNA.” cannot be quickly drawn from figure 3. The miRNAs in (a) and genes in (b) did not match well. I did not find the miRNA for the gene Citrus_au_3104 in (a). Please carefully modify the figure.
2, The scions of Mexican lime were grafted on C. macrophylla. Both of the citrus varieties are susceptible to C. Las infection. I would like to know if the rootstock affects the Mexican lime scion, which might interfere with the miRNA profiles during the citrus-C. Las interaction.
Minor
1, Please provide the scientific name for Mexican lime in the abstract.
2, Please pay attention to the grammar mistakes in the manuscript. I give a few examples.
Line 22, “knowledge derived from not model systems,” “not model” is inappropriate.
Line 54, “C.Las induced” should be “C.Las-induced”.
Line 56, “transcriptional repression was mainly observed in redox, photosynthesis and cell wall” This sentence is incomplete.
Author Response
"Please see the attachment"

Round 2
Reviewer 1 Report
This Is a folowup review of "Differential expression of genuine miRNAs Involved In response to Candidatus Liberibacter asiaticus Infection In Mexican lime at early and late stages of Huanglongbing disease.
Point 4: Title) “genuine” suggest that there are other options. Line 67-68 indicates many are doubtful and some are authentic. The problem appears to be false positives in the algorithms used to identify miRNAs.
Response 4: The use of the word genuine is derived from the same paper by Michael Axtell and Blake Meyers, "Revisiting Criteria for Plant MicroRNA Annotation in the Era of Big Data". In that paper, the authors express their frustration with the presence in miRBase of "miRNAs" that are not actual miRNAs. We use the word "genuine" to convey the idea that the miRNAs that we identified were done so using ShortStack, a small RNA sequencing-based method and the best option currently available.
Point 4a: Yes, but using “the most accurate tool” is no guarantee of accuracy. “Genuine” is overselling an outcome, and just because others do it does not make it good practice. Using “genuine” as a short form of “we used ShortStack software” is a bad practice. I would suggest removing genuine.
Line 163) while the author’s response to the reviewer indicated that “Plant small RNA genes” is the name of a specific database, that is still not clear in the text. The database is not properly cited.
Response 15: Excellent response, but I could not find it in the manuscript. It needs a little rewording to be appropriate, but it tells the reader that you are defining “early” and “late” in a specific way based on your protocols for studying the pathogen. It also warns readers not as familiar with the system that other locations may have to redefine “early” and “late” to match disease progression in their area.
Response 18:
10 infected plants with CLas (HLB+) and
10 healthy plants as control (HLB-) were selected.
At this point I have 20 plants in total, ten In each treatment (HLB+ and HLB- respectively).
For each sample,
Sample of what?
five individual plants were selected according to homogeneity on the disease symptoms.
Out of the ten plants in each treatment, half (5) were selected based on symptom expression (the other 5 were destroyed?)
A total of 8 pooled samples were generated.
8 pooled samples? Are you sampling leaves, stems, roots? If I collect one sample from each plant I will have five samples. I can get eight samples by collecting two samples from three plants and one sample from the other two, or a variety of other possibilities. Is this 8 pooled samples (what was pooled) or 8 samples that are pooled into one sample for processing? Would you not have 4 leaves pooled into one sample, not 8?
The text as written makes little sense and needs additional detail so that a reader can figure out what was done. Details can be left out where the details are filled in by the cited work. However, readers should not be confused by what is written even if they do not choose to look up the cited article.
In the cited work 45 lime plants were inoculated HLB+, and 15 were mock inoculated (HLB-). After inoculation, samples were collected at 8 and 16 weeks. All plants were sampled by collecting 8 leaves from each plant (see Arce-Leal et al 2020 for details). Four of the leaves were ground in liquid nitrogen for RNA-Seq analysis, and the other four leaves had midribs and petioles removed for qPCR analysis. A total of eight libraries were generated. Did you mean a total of eight libraries per plant, or eight in total for the entire experiment? For each library, 5 individual plants … why sample 45 plants only to select five? After extraction (grinding in liquid nitrogen) how did you evaluate disease symptom expression. Please write methods in a set of steps that can be used: I grow plants -> I select plants -> I treat or process plants -> I analyze samples -> I draw conclusions from the analysis. To generate libraries, I must have already collected samples. Selecting plants after gathering samples is not a typical approach. Write methods as a process in the order that each step was performed.
Line 384: You start with ten plants in a treatment. You select the best five. You take eight leaves per plant and homogenize four of them for RNA-Seq, and the midribs and petioles from the other four are homogenized for qPCR. This is now one sample, yet you claim 8.
No indication how the samples were segregated into two biological replicates.
Make this clearer! The cited work helps a little, but also leaves more questions.
Line 278) If I plant a Mexican lime seed I will not have a tree with 8 tertiary branches 11 months later. Depending on how I did things, I might have such a plant if I grafted larger branches onto older rootstock such that I used plants that were grafted 9 months ago. Are details missing?
Line 379: How was the screen-house maintained pathogen free? The screen mesh size small enough to prevent fungal or bacterial spores from penetrating would have serious consequences on light levels. Most structures cannot be maintained pathogen free without special precautions: double doors, air filters, wind screens (blow pests off people), sprays on people and equipment, and so forth. Such extreme measures are uncommon. It would also require clean plants. How were they kept clean? Given clean plants, a restricted entry greenhouse, and a few other precautions I can see maintaining a pest free screen-house without use of pesticides. However, such care is not evident in the methods.
While the authors claim to have added details about the growing conditions of their plants, they did not provide enough detail. Furthermore the article cited for additional methods is also missing details. Plant response to disease is influenced by plant health which is in turn a function of soil, nutrients, water, light, and other factors. I do not care if you used soil or growing media, but I do care that readers know what soil type was used or the brand name and mix of a soilless growing media.
Author Response
"Please see the attachment"
